# The Potential Compatibility of Designating Offshore Wind Farms within Wider Marine Protected Areas—Conservation of the Chinese White Dolphin Regarding Fishers' Perception

Hao-Tang Jhan, Hsin-Ta Lee and Kuo-Huan Ting *

Institute of Marine Affairs and Business Management, National Kaohsiung University of Science and Technology, Kaohsiung 811213, Taiwan
* Correspondence: dgh0809@nkust.edu.tw

**Abstract:** The population of the Chinese white dolphin along Taiwan's west coast is under a range of threats. The designation of marine protected areas (MPA) is urgently required for their protection. However, conflicts between specific species conservation and fishing rights mean that the success of such a designation relies on the fishers' perceptions and awareness of an MPA. Designating offshore wind farms within MPAs can be a mechanism for minimizing conflicts between fisheries and conservation. The purpose of this study is to examine the potential for designating an offshore wind farm within an MPA for Chinese white dolphin conservation by exploring the attitudes of local fishers. This study used face-to-face questionnaires. The results show that the main challenges are conflicts of interest, insufficient science-based information, and inadequate law enforcement. Offshore wind farms could be a way to maximize the benefits for different stakeholders and positively impact the marine environment and ecosystem. This study makes feasible recommendations on how to improve conservation, promote renewable energy, and encourage sustainable fisheries.

**Keywords:** Chinese white dolphin; marine protected area (MPA); offshore wind farm; stakeholder





## 1. Introduction

Under the threat of climate change, green energy development has become a global trend. The geography of the Taiwan Strait, with its prevalent northeast and southwest monsoons, makes it a suitable and cost-effective location for the development of offshore wind farms [1]. Since 2012, the Taiwan government has actively promoted the "Thousand Wind Turbines" project under which more than 1000 wind turbines (800 offshore and 450 onshore) will be constructed by 2030, with a combined rated capacity of 5.2 GW [2–4]. However, although offshore wind farms are environmentally friendly and carry low risks, they can still cause harm to the local environment and communities during their installation, operation, and decommissioning [5]. This includes the overlapping of the offshore wind farms with traditional fishing grounds, natural habitat destruction, fragmentation, nuisance, and displacement [6]. Moreover, several planned offshore wind farms off the west coast of Taiwan are close to the wild habitat of the endangered Indo-Pacific humpback dolphin or Chinese white dolphin (*Sousa chinensis*).

This species is observed in coastal waters from southeast Asia to northern Australia [7–9]. Since 1994, its population has been classified as Critically Endangered (CR) by the International Union for Conservation of Nature and Natural Resources [10]. The number of Chinese white dolphins in the Eastern Taiwan Strait is approximately 100 and continues to decline [11,12]. They live along the west coast and estuaries of Miaoli, Taichung, Changhua, and Yunlin counties, about three kilometers offshore, where human activity is frequent. This includes ports and industrial and aquacultural areas [13]. As a result, the population of the Chinese white dolphin along this coast is under a range of severe threats, including:

- **habitat degradation or loss:** 80% of the western coast is artificial, with industrial development, fishing ports, offshore wind farms, and land reclamation having caused critical degradation or loss of the Chinese white dolphin's natural habitats during both construction and operation phases [12,14–25].
- **underwater noise:** The National Marine Fisheries Service (NMFS) determined that sound pressure is likely to cause physical harm and behavioral disturbance. Impulsive sound sources (e.g., impact pile driving, air guns) and continuous sound sources (e.g., vibratory pile driving and mechanical dismantling) during the construction of offshore wind farms may cause physical injury to the auditory senses of cetaceans [12,15–18,20–25].
- **pollution:** industry, agriculture, and households all contribute to wastewater that is released into the ecosystem. Although it is difficult to prove a relevant link between the death of Chinese white dolphins and the accumulation of heavy metals in the environment, water pollution can impact their reproductive and immune systems and their prey resources [16,17,21–25].
- **fishing activity:** incidental catch from trawling and gillnet fisheries is the greatest threat to the Chinese white dolphin in this area, leading to injury and death. Further, overfishing and illegal fishing have resulted in prey reduction and habitat degradation, which threaten the population of Chinese white dolphins [12,16,17,21–25].

Over the past few decades, the main purpose of large-scale development projects along the west coast of Taiwan has been to stimulate the economy and create more job opportunities. In recent years, the public awareness of environmental protection has increased, and, therefore, such coastal developments have given greater attention to the local environment and ecosystems. Marine protected areas (MPAs) are widely regarded as the most effective tools for maintaining the stability and resilience of marine environments and ecosystems [26]. They play a significant role in biodiversity and marine resource conservation [27,28]. In order to conserve the population of the Chinese white dolphin, a range of wildlife habitats have been specially designated as MPAs in Asia. These include Xiamen and Zhuhai in China and Sha Chau and Lung Kwu Chau Marine Park in Hong Kong [29–33]. However, along Taiwan's western coast, the activity and habitat range of the Chinese white dolphin overlaps with local fishing grounds, leading to conflicts between species conservation and fishing rights. Fishers' perceptions and awareness are thus critical to the success of any MPA designation.

The development of offshore wind farms may be a mechanism for minimizing conflicts between fisheries and conservation. Although the negative impact of offshore wind farms on the marine environment and creatures is well known, the "reef effect" may also bring positive benefits for certain marine species through the increase in habitat complexity, biodiversity, and abundance [34–38]. Operational offshore wind farms within MPAs can restrict fishery activities, preventing environmentally damaging activity around turbines and cables. More specifically, offshore wind farms can directly or indirectly play this role for the Chinese white dolphin, ameliorating the conflicts between ecological conservation and commercial fisheries through a fisheries compensation and livelihood restoration plan (LRP) from wind farm project developers. Offshore wind farms, therefore, appear to bring an opportunity for the development of green energy, ecological conservation, and the provision of local economic benefits. The purpose of this study is to explore the attitude and perception of fishers towards these issues, with the aim of defining the potential compatibility of designating offshore wind farms within MPAs.

## 2. Research Method

In successfully establishing an MPA for the Chinese white dolphin, the fishers' attitudes toward such a designation is a key factor. Therefore, the study area covered the four coastal townships of Mailiao, Taixi, Sihu, and Kouhu, close to the natural habitats of the Chinese white dolphin and the planned location of an offshore wind farm in Yunlin County. Information was collected from local coastal and offshore fishers by a face-to-face

questionnaire. This approach was acceptable to all interviewees who completed the survey. The broad coverage and dedicated focus on the intended population was a key advantage of this approach and returned a higher response rate than other methods (mail, telephone, and online). To ensure this study contacted those local offshore and coastal fishermen with sufficient experience to provide useful information, snowball and purposeful sampling methods were employed. All of the interviewees fished within the waters of the habitat of the Chinese white dolphin. Statistical Product and Service Solutions (SPSS) ver. 20.0 was used to summarize and present the results. An independent sample t-test was used to explore the differences in fishers' opinions regarding the conservation of the Chinese white dolphin depending on whether they support or oppose the development of offshore wind farms. The questionnaire consisted of 42 entries and was divided into 4 parts, as follows:

- Information on the fisher and fishing activities (personal information, fishing vessel and operation)
- Fisher's perceptions of the Chinese white dolphin (population, interaction, impacts)
- Fisher's perceptions of marine protected areas (understanding, institutional trust, incentives)
- Adaptive actions and available options (conservation options, available resources)

**3. Results**

The questionnaire survey was conducted over two months (from August to September) in 2020. Since most fishers were elderly and with low literacy, the interview process was conducted orally, with the responses being written down by the interviewers. This also proved advantageous in improving the valid response rate and the quality of the data. A total of 69 local offshore and coastal fishers voluntarily participated in this survey, giving a high overall survey response rate (76.7%). Table 1 shows the demographic data of the respondents and their fishing activities. It reveals that the ratio of men was relatively high. About 60% of the participants had at least 20 years of fishing experience. These figures are consistent with a general and severe aging trend found within the fishery sector in Taiwan. PVP raft and gillnet are, respectively, the major fishing vessel and the most common fishing techniques used in offshore and coastal fisheries of Yunlin County.

**Table 1.** Demographic information of the respondents (*n* = 69).

| Items | | Frequency | Percentage (%) |
|---|---|---|---|
| Gender | Male | 55 | 79.7 |
| | Female | 14 | 20.3 |
| Age | <20 years old | 0 | 0 |
| | 20–30 years old | 2 | 2.9 |
| | 30–40 years old | 7 | 10.1 |
| | 40–50 years old | 13 | 18.8 |
| | 50–60 years old | 19 | 27.5 |
| | >60 years old | 28 | 40.6 |
| Experience | <5 years | 7 | 10.1 |
| | 5–10 years | 7 | 10.1 |
| | 10–15 years | 8 | 11.6 |
| | 15–20 years | 6 | 8.7 |
| | 20–25 years | 6 | 8.7 |
| | >25 years | 35 | 50.7 |
| Educational background | Self-study | 5 | 7.2 |
| | Elementary school | 24 | 34.8 |
| | Junior high school | 13 | 18.8 |
| | Senior high school | 19 | 27.5 |
| | Bachelor's degree | 8 | 11.6 |

**Table 1.** *Cont.*

| Items | | Frequency | Percentage (%) |
|---|---|---|---|
| Vessel types | Sampan | 5 | 7.2 |
| | PVP (polyvinylpyrrolidone) raft | 58 | 84.1 |
| | Tonnage ≤5 | 5 | 7.2 |
| | Tonnage 5–9 | 0 | 0 |
| | Tonnage 10–19 | 1 | 1.4 |
| Fishing technique | Pole and line | 2 | 2.9 |
| | Gillnet | 67 | 97.1 |

*3.1. Fishers' Experience of the Chinese White Dolphin*

Approximately 60% of the participants stated that they believe the population of the Chinese white dolphin has declined. More than half think that their catch is lower when the Chinese white dolphins are nearby their fishing grounds. Most stated that there is a competitive relationship between both the fishers and Chinese white dolphins for fishery resources. Nevertheless, most local fishers agree that habitat destruction, pollution, and climate change have also had a significant impact on their landings. Table 2 also shows that the Chinese white dolphin is most frequently encountered by fishers from April to September. This is in complete agreement with [12]. Since Chinese white dolphin numbers are low, it is unsurprising that only six gillnet fishers have ever incidentally caught them.

**Table 2.** Fishers' experience of the Chinese white dolphin (*n* = 69).

| Items | | Frequency | Percentage (%) |
|---|---|---|---|
| Change in the Chinese white dolphin population from 10 years ago | increase a lot | 0 | 0 |
| | increase slightly | 14 | 20.3 |
| | no change | 15 | 21.7 |
| | decrease slightly | 25 | 36.2 |
| | decrease a lot | 15 | 21.7 |
| Impact on catches when the Chinese white dolphin is nearby fishing grounds | increase a lot | 2 | 2.9 |
| | increase slightly | 19 | 27.5 |
| | no change | 10 | 14.5 |
| | decrease slightly | 27 | 39.1 |
| | decrease a lot | 11 | 15.9 |
| Frequency of spotting Chinese white dolphins | Jan–Mar | 5 | 7 |
| | Apr–Jun | 27 | 39.3 |
| | Jul–Sep | 22 | 31.6 |
| | Oct–Dec | 3 | 5 |
| | No difference | 12 | 17.1 |
| Incidental catch of Chinese white dolphin | yes | 8 | 11.6 |
| | no | 61 | 88.4 |

*3.2. Fishers' Opinions on the Conservation of the Chinese White Dolphin*

It is notable that 68.1% of the respondents support the conservation of the Chinese white dolphin. However, their primary reason is to protect the marine environment and conserve marine resources. Consequently, only 23.2% of the respondents were willing to participate in the conservation work. Although 56.5% of the participants agreed that the conservation of the Chinese white dolphin is important for the marine ecosystem, the figures for its contribution to fishery resource preservation and fishery income are lower, 44.9% and 31.8%, respectively (Table 3). These results reveal that there is considerable public concern about the potential negative impact of conservation actions on local fishers' incomes and livelihoods.

**Table 3.** Fishers' opinions on the conservation of the Chinese white dolphin (*n* = 69).

| Items | | Frequency | Percentage (%) |
|---|---|---|---|
| Support the conservation of the Chinese white dolphin? | strongly agree | 8 | 11.6 |
| | agree | 39 | 56.5 |
| | neutral | 15 | 21.7 |
| | disagree | 7 | 10.1 |
| | strongly disagree | 0 | 0 |
| Willing to participate in conservation work for the Chinese white dolphin | strongly agree | 0 | 0 |
| | agree | 16 | 23.2 |
| | neutral | 10 | 14.5 |
| | disagree | 38 | 55.1 |
| | strongly disagree | 5 | 7.2 |
| The conservation of the Chinese white dolphin is important for the marine ecosystem | strongly agree | 6 | 8.7 |
| | agree | 33 | 47.8 |
| | neutral | 17 | 24.6 |
| | disagree | 12 | 17.4 |
| | strongly disagree | 1 | 1.4 |
| The conservation of the Chinese white dolphin can contribute to fishery resource preservation | strongly agree | 5 | 7.2 |
| | agree | 26 | 37.7 |
| | neutral | 24 | 34.8 |
| | disagree | 14 | 20.3 |
| | strongly disagree | 0 | 0 |
| The conservation of the Chinese white dolphin can contribute to fishery income | strongly agree | 5 | 7.2 |
| | agree | 17 | 24.6 |
| | neutral | 27 | 39.1 |
| | disagree | 20 | 29 |
| | strongly disagree | 0 | 0 |

*3.3. Fishers' Attitudes towards the MPA for the Chinese White Dolphin*

Table 4 shows that only 34.8% of respondents support the establishment of an MPA for the Chinese white dolphin. Even if fishing activities were allowed within the MPA, just 37.6% of the respondents approved. These findings can be attributed to the fishers' doubts about the potential contributions of the MPA to fishery landings and income and the impact of fishing gear restrictions in the MPA area.

**Table 4.** Fishers' attitude towards the MPA for the Chinese white dolphin (*n* = 69).

| Items | | Frequency | Percentage (%) |
|---|---|---|---|
| Support for the MPA in the surrounding waters | strongly agree | 2 | 2.9 |
| | agree | 22 | 31.9 |
| | neutral | 12 | 17.4 |
| | disagree | 27 | 39.1 |
| | strongly disagree | 6 | 8.7 |
| Support for the MPA in the surrounding waters if fishing activities are allowed in specific areas | strongly agree | 1 | 1.4 |
| | agree | 25 | 36.2 |
| | neutral | 7 | 10.1 |
| | disagree | 28 | 40.6 |
| | strongly disagree | 8 | 11.6 |
| The MPA will contribute to loadings | strongly agree | 3 | 4.3 |
| | agree | 17 | 24.6 |
| | neutral | 19 | 26.1 |
| | disagree | 26 | 37.7 |
| | strongly disagree | 5 | 7.2 |

**Table 4.** *Cont.*

| Items | | Frequency | Percentage (%) |
|---|---|---|---|
| The MPA will contribute to fishery income | strongly agree | 3 | 4.3 |
| | agree | 16 | 23.2 |
| | neutral | 7 | 10.1 |
| | disagree | 37 | 53.6 |
| | strongly disagree | 6 | 8.7 |
| Gillnet fishing should be restricted in the MPA | strongly agree | 1 | 1.4 |
| | agree | 17 | 24.6 |
| | neutral | 18 | 26.1 |
| | disagree | 30 | 43.5 |
| | strongly disagree | 3 | 4.3 |

Table 5 shows that most respondents (70%) were willing to cooperate with various fishery management strategies and restrictions. It reveals that 28.9% approved of a seasonal fishery closure, while roughly one quarter supported other fishery management measures, such as closed fishing grounds (13.3%) and the restriction of specific fishing gear (13.3%). Not surprisingly, 30% of fishers were unwilling to accept any kind of management measure or restriction. This clearly demonstrates that the majority of local fishers are willing to cooperate with one or another fishery management measure providing they achieve the objective of fishery resources restoration.

**Table 5.** Fishers' support for fishery management measures and restrictions in the MPA for the Chinese white dolphin ($n = 69$).

| Fishery Management and Restrictions | Frequency | Percentage (%) |
|---|---|---|
| Closed fishing ground | 12 | 13.3 |
| Closed fishing seasons | 26 | 28.9 |
| Total allowable catch (TAC) | 1 | 1.1 |
| Restriction of fishing gear | 12 | 13.3 |
| Changing fishing time | 5 | 5.6 |
| Changing fishing ground | 7 | 7.8 |
| I don't support the above management actions | 27 | 30.0 |

*3.4. The Effects of the Development of Offshore Wind Farms on the Fishers' Opinions*

Table 6 shows the independent sample t-test results. It indicates differences in fishers' opinions depending on whether they support or oppose the development of offshore wind farms. Along the west coast of Taiwan, there is considerable overlap between fishing grounds, the occurrence of Chinese white dolphin and its habitat, and the location of the offshore wind farms. The t-test indicates a significant difference between supporters and opponents in the willingness to participate in the conservation actions for the Chinese white dolphin ($p = 0.02 < 0.05$). The mean number of fishers who supported the development of offshore wind farms and were willing to participate in conservation actions was higher (2.88) than that of those who opposed the development (2.34). There was also a significant difference when it came to the engagement of fishers in the discussion process prior to the establishment of the MPA ($p = 0.04 < 0.05$). However, there was no significant difference in either support for the MPA for the Chinese white dolphin in the surrounding waters ($p = 0.391 > 0.05$) or the acceptance of restrictions on gillnet fishing ($p = 0.195 > 0.05$). There was general agreement that fishers' opinions played a critical role in the discussion and decision-making. However, the lack of a convenient time for fishers to participate in the discussion process and distrust in management institutions may have reduced their willingness to participate in the planning process.

**Table 6.** Statistical testing on fishers' opinions regarding the conservation of the Chinese white dolphin (*n* = 69).

| Test | Items | Mean | | t Value | Associated *p*-Value |
|------|-------|------|---|---------|----------------------|
| | | **Supporters** | **Opponents** | | |
| 1 | Support an MPA for the Chinese white dolphin in the surrounding waters. | 2.96 | 2.73 | 0.863 | *p* = 0.391 > 0.05 |
| 2 | Restrict gillnet fishery in an MPA for the Chinese white dolphin. | 2.56 | 2.86 | −1.310 | *p* = 0.195 > 0.05 |
| 3 | Willing to participate in conservation actions for the Chinese white dolphin. | 2.88 | 2.34 | 2.386 | *p* = 0.02 < 0.05 |
| 4 | Agree that most fishers will comply with the conservation measures of an MPA. | 2.76 | 2.84 | −0.324 | *p* = 0.747 > 0.05 |
| 5 | Agree that fishers should participate in the discussion process more often before establishing an MPA. | 4.16 | 3.70 | 2.094 | *p* = 0.04 < 0.05 |
| 6 | Agree that consultation with fishers must occur before the decision to establish an MPA is made. | 4.36 | 4.07 | 1.785 | *p* = 0.079 > 0.05 |
| 7 | Willing to participate in the public hearing related to the planning of an MPA for the Chinese white dolphin. | 4.12 | 3.82 | 1.379 | *p* = 0.173 > 0.05 |

## 4. Discussion

In 2020, Taiwan's Ocean Affairs Council established a Major Wildlife Habitat (MWH) for the Indo-Pacific Humpback Dolphin. Covering an area of 763 square km, it combines marine and estuarine ecosystems. However, although the MWH designation is an important step forward, it is more symbolic than substantive, as legal fishing activities in the original application may continue in the area, and other development has been only minimally restricted. The purpose of the MWH is to monitor and control development, but it lacks sufficient active conservation and management programs. In order to protect the Chinese white dolphin in the shallow waters off Taiwan's west coast, an MPA is urgently needed. However, the major challenge facing MPA designation is how to strike a balance between economic concerns and environmental sustainability.

### 4.1. Conflicts between Fisheries and MPAs

The fact that, despite supporting the conservation of the Chinese white dolphin, most users still oppose the MPA designation can be attributed to the substantial overlap between the MPA, the exclusive fishing zone, and the fishing grounds of local fishers. Fishers are clearly concerned that their livelihood and the local economy will suffer if their fishing rights are excluded from the MPA. It is well known that the Chinese white dolphin and local fishers are competitors for fish resources and that incidental catch is a critical threat to the population of the Chinese white dolphin. However, the majority of respondents in our survey claim to have a very low incidental catch rate, and cetaceans are not their target. This suggests that local fishing activities are not the major cause of the decline in the number of Chinese white dolphins.

In order to deal with the various conflicts among stakeholders in a multiple-use MPA, zoning is a key management tool in the development of management objectives and strategies. For example, the MPAs for cetaceans in China [30] and Canada [39] were designed to allow particular fishing activities in designated zones. The Chinese white dolphin's habitats related to ecological factors, including breeding, recruitment and nursery grounds, fish aggregation sites, current patterns, and population stability, can be designated

as a core zone. A competent authority can then institute management measures and actions to strictly exclude illegal fishing and development in this zone. The remaining areas of the core zones can be designated for the sustainable use of fish resources exclusively by the local community and fishers. MPA zoning is a planning tool to enable the recovery of fish resources for both the Chinese white dolphin and local fishers and to promote a healthy, resilient, and diverse marine ecosystem [40,41]. Ref. [42] states that neglecting social and economic issues when planning an MPA reduces the likelihood of its success. Our research results support the claim that management measures involving fish resource conservation are more acceptable to local fishers. These include a closed fishing season (28.9%), a no-take zone (13.3%), and restrictions on allowed fishing gear (13.3%). Therefore, stakeholders' (fishers and village leaders) participation is required to ensure successful and equitable management outcomes [43–45]. By incorporating the experiences and preferences of stakeholders in the planning process, MPA management strategies can be improved.

### 4.2. Insufficient Science-Based Information and Inadequate Law Enforcement

One major challenge highlighted by local fishers is the lack of science-based data on the biology and life cycle of the Chinese white dolphin. They say that they need accessible and acceptable information or data to convince them of the appropriateness of the location and size of the MPA. For this reason, an MWH designation may be more suitable for the natural habitats of Chinese white dolphins. This also corresponds to the precautionary principle for achieving sustainable governance of the oceans [46–48]. Although there is insufficient science-based information available to determine the most effective management strategy for the Chinese white dolphin, small-scale adaptive steps can be taken in the area with less conflict in order to gradually improve the effectiveness of conservation measures [21,49].

Local fishers attribute the fall in the number of Chinese white dolphins and fishery resources to pollution and illegal fishing. The decline in coastal fishery resources and the marine environment may also be due to inadequate enforcement of current regulations. The Ocean Conservation Act has not yet been approved in Taiwan; however, the Wildlife Conservation Act, Fisheries Law, and Tourism Development Act have sufficient legal provisions for the development of an MPA. If these existing acts and regulations can effectively restrict industrial pollution and specific fishing gear and methods, it may be possible to sustain the population of the Chinese white dolphin. However, respondents realize that law enforcement is inadequate and management effectiveness is low and it is difficult to guarantee that existing fishing activities may continue after the MPA designation. Sha Chau and Lung Kwu Chau Marine Park in Hong Kong issued fishing licenses to allow local fishers or residents to retain their fishing rights in the marine park [29,50,51]. This system strictly controls illegal fishing activities and cross-border trespassing in order to protect the fish resources and marine environment in the reserved area. The integration of cross-department and systematic planning is necessary in terms of jurisdiction and spatial planning to ensure that management actions and law enforcement are effective in an MPA [52].

### 4.3. Designating Offshore Wind Farms as MPAs

Previous studies have indicated the potential for designating an offshore wind farm as a no-fishing zone or a restricted activity zone within a wider MPA [45,53–55]. The Spanish government carried out a strategic environmental assessment of an offshore wind farm, incorporating a newly designated marine protection area [34]. Spanish waters were divided into three zones according to their suitability for an offshore wind farm. These zones were suitable, suitable with conditions and exclusion zones. While the purpose of an offshore wind farm and MPA differs, their functions may be complementary in certain circumstances. The designation of an MPA in the vicinity of an offshore wind farm would be likely to benefit multiple stakeholders [56].

The development of offshore wind farms within wider MPAs is advancing at a rapid pace. If improperly implemented, it can rob fishers and communities of use, control, or

access to resources [57]. This may reduce the available fishing grounds and displace fishing efforts into other unprotected or undeveloped areas [45,58]. Thus, the cumulative environmental impact may increase because of the presence of an offshore wind farm in a closed area and because of increased fishing efforts in open areas [59]. It will also have social and economic consequences since fishing provides not only traditional employment opportunities but also cultural identity and tourist interest in many areas where offshore wind farms are developed [60,61]. Although an offshore wind farm is environmentally friendly and poses a lower risk, it is not totally harmless to the marine environment and ecosystem, particularly during the installation, operation, and decommissioning stages [5]. Many studies have highlighted the range of generic threats to the local environment, including habitat loss, collision and entanglement, noise, and electromagnetic fields [18,62–65]. The actual impact of these threats will vary significantly between the different stages [5].

On the other hand, there is increasing evidence to suggest that, with appropriate design, siting, and management, offshore wind farms may actually produce a positive environmental impact, for example, as an artificial reef, a fish aggregation device, and through spatial restriction [35,36,38,66–69]. As artificial reefs, offshore wind farm structures may also have positive effects on specific marine species as they create new habitats and restrict trawling and gillnet operations [45]. In addition, with a higher survival rate of marine organisms and the appearance of bigger individuals within the boundaries, spillover to the surrounding areas can be expected [69]. Ref. [70] suggested that this spillover effect could mitigate the negative impact of access loss or fishery restrictions around the offshore wind farm because it would increase the proportion of high trophic level species. It also shows that the expected increase in biomass and catches is highly localized in the exclusion zone, which could, therefore, play the role of a fish aggregating device by attracting predators from the surrounding areas.

Incorporating a no-take zone around an offshore wind farm within a wider MPA may maximize the benefits by increasing the abundance and occurrence of economically important species that support commercial and recreational fishing, as well as associated local fishing [71,72]. Closing the area near underwater obstructions may also increase local fishery yields by protecting juvenile fish. Due to the risk of entanglement in the anchoring structures of wind turbines and the potential for gear damage or loss, it is likely that fishers will be hesitant to deploy long lines, gillnets, and trawls in the area [73,74]. In one instance, an oil platform has acted as an exclusive fishing zone since the platform structure prevents the use of various types of commercial fishing gear [75].

Fayram and De Risi [56] have suggested that excluding or limiting fishing activities in areas surrounding an offshore wind farm may help to control fishing efforts and total harvest. In this way, local fishers may benefit from increased yields, particularly since overfishing is a serious problem in Taiwan. Likewise, local fishers and fishery managers will benefit because the total harvest can be controlled, and the fishery can be sustainably managed. The owner of the offshore wind farm will also benefit because the wind turbine is less likely to be damaged by fishing activities. Developers of offshore wind farms have also come up with the innovative idea of installing fish farms around the perimeter of wind farms in order to increase benefits for multiple stakeholders [34,76].

## 5. Conclusions and Policy Implications

The conservation of the Chinese white dolphin and economic development are important to the local community. Economic incentives associated with conservation may strengthen the support and participation of the local community, especially with regard to the issue of the sustainable development of the local industry.

Although Taiwan has already designated an MWH, conflicts and doubts remain when it comes to designating an MPA. While it may seem like a panacea, the efficacy of an MPA will depend on a number of environmental and ecosystem variables within the designated area. In this respect, the limited available evidence highlights the need for further research involving long-term monitoring at different sites to ensure better and more acceptable

conservation options. Further, clear evidence for the actual effects of magnetic fields, electromagnetic fields, and anthropogenic underwater noise on marine mammals during the construction and operational phases remains very poor. More research is needed to determine the potential for chronic and long-term effects and also to develop alternative engineering techniques to eliminate these impacts. Effective monitoring both before and after the construction of offshore wind farms is essential in determining the successful recovery of the Chinese white dolphin population and the positive benefits to local fisheries. Considering the further expansion of the global offshore wind farm industry, developers and managers will benefit from increased experience and knowledge [77]. In view of existing practices in Taiwan, this research proposes the following:

- Develop ecotourism and green energy education tourism to improve local economies

As a number of respondents are concerned that MPA designation may affect their livelihoods, dolphin-watching tours and charter fishing can be established as an alternative source of income. Such marine ecotourism can combine the Chinese white dolphin with 'green' education about wetland resources and offshore wind farms. This can attract the participation of local communities and reduce the ecological impacts on the marine environment.

- Carry out MPA zoning management and science-based investigations

The MWH for the Chinese white dolphin overlaps the main fishing grounds of local offshore and coastal fishers. The designation of an MPA without appropriate countermeasures may affect their livelihood and increase their opposition to the development. Zoning can reduce such conflicts by creating an inclusive, coordinated, and comprehensive scheme for MPAs and other ocean uses. Designating an offshore wind farm as an MPA or fishing exclusion zone in Taiwan is a feasible approach that could benefit all stakeholders involved. However, the benefits and degree of compatibility between the offshore wind farm, MPA, and local fishery will depend on the fishing gear, fish species, and location. Therefore, it is vital to gather more information on habitat utilization and ecosystems. Long-term science-based investigations and data collection on Chinese white dolphin food sources, the influence of fishing on stock, and other related issues are necessary to clarify the activity range and the physiological status of the Chinese white dolphin. Coordination of conservation measures and offshore wind farm development can be facilitated through enhanced information exchange among authorities [77].

- Encourage public participation and information exchange during the planning stage

Public participation and community engagement are fundamental for a successful MPA designation. This is particularly true in Taiwan, which lacks experience in designating MPAs for cetaceans. A dialogue between government, stakeholders, and the public is, therefore, crucial. However, there may be a number of significant constraints and barriers that limit such community engagement. These include time constraints, inadequate information and communication, and a lack of confidence in government information and policies. The implementation of channels of communication before designation would help the local community to make observations and suggestions, thereby reducing the potential for controversy within the designated ranges and facilitating better conservation work.

**Author Contributions:** Conceptualization, H.-T.J. and K.-H.T.; methodology, H.-T.J. and K.-H.T.; validation, H.-T.J. and K.-H.T.; formal analysis, H.-T.L.; investigation, H.-T.J. and H.-T.L.; resources, H.-T.J. and H.-T.L.; data curation, H.-T.L.; writing—original draft preparation, H.-T.J.; writing—review and editing, K.-H.T.; supervision, H.-T.J. All authors have read and agreed to the published version of the manuscript.

**Funding:** This research was supported by Yunneng Wind Power Co., Ltd., Taiwan.

**Data Availability Statement:** The sources of data supporting this study are illustrated in the report.

**Acknowledgments:** The authors acknowledge the contribution of the people from the Yunlin Fisherman's Association, local fishers, and the community for providing valuable information and assistance.

**Conflicts of Interest:** The authors declare no conflict of interest.

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
