# Peer review of "The Potential Compatibility of Designating Offshore Wind Farms within Wider Marine Protected Areas—Conservation of the Chinese White Dolphin Regarding Fishers’ Perception"

_fishes, doi:10.3390/fishes7040195_

Round 1
Reviewer 1 Report
The manuscript is generally well organized, the field work is adequately described, and the references adequate - even if some additional references are needed at certain passages, see below -. My main concern regard the focus of the whole work, which is positively unbalanced toward one out of the several possible outcomes of a possible spatial scheme of management different uses of the sea within the study area, namely the protection provided by a wind-mill farm to an endangered dolphin species, in such a way that the wind-mill farm would act "as" a marine protected area. As stated, such idea is incorrect, since the specific policy-oriented goals, operational objectives, mechanisms of governance, and expected outcomes of marine protected areas and wind-mill farms are completely different. It is true that wind-mills represent physical obstacles to some tipes of navigation, and structures deployed onto the sea bottom preclude some types of fishing. However, it is also true that wind-mills generate underwater noise and electromangnetic fields for which we lack knowledge about potential long-term impacts on dolphins in general, and on the focused dolphin species in particular. Authors emphasize the expected positive outcomes, while ignoring or briefly mentioning those actually or potentially negative. Another example regards the well known "positive effect"of artificial structures at sea, as they became populated by many diverse organisms in an otherwise "empty and monotonous" extensive soft bottom. This idea is present in scientific literature since the '70s, and it is contestated because artificial structures also functions as stepping stones for invasion of marine exotic (or alien) species of flora and fauna, attract pradators that insensively feed on the bottom fauna generating "defaunation halos" around the artificial structures, attract large fishes that are easily found by both professional and recreational fishermen thereby increasing fishinhg mortality, etc... My message is that a sound scientrific work must recall the different views, evidences, and positions regarding any discussed option, in order to unbiasedly inform the scientific community and, sucessively, the involved stakeholders and the wide public. Presenting only the positive outcome is useful but lacks rigour.
Author Response
The authors greatly appreciate the thoughtful comments and suggestions.
- The revised manuscript has been carefully revised by two highly qualified native English speakers to improve the grammar, punctuation, spelling, and overall style.
- Thank you for this valuable feedback. Our research focuses on the potential of designating offshore wind farms as MPAs because the establishment of MPA is very difficult in Taiwan. All authors agree that the purpose and function are very different between wind farms and MPAs, but offshore wind farms can also produce positive environmental impacts with appropriate design, siting and management. This article also explains the possible risks to marine environment and ecosystem and socioeconomic consequences in section 4.3. We agree that the knowledge about potential long-term impacts on dolphins is insufficient, and therefore the authors suggest that a long-term science-based investigation is necessary to clarify the impacts of offshore wind farm to marine environment and ecosystem in the discussion and conclusion section.
Reviewer 2 Report
This manuscript requires a complete review for grammar and English. I attach some of the corrections I suggest to line 199. This took me 2days to do.

Author Response
The authors greatly appreciate the thoughtful comments and suggestions.
- The revised manuscript has been carefully revised by two highly qualified native English speakers to improve the grammar, punctuation, spelling, and overall style.
- PVP raft is a kind of raft made by PVP (Polyvinylpyrrolidone) tubes, PVP raft is very common in Taiwan's offshore and coastal fisheries. (Line 138)
- 3.2 Fishers' opinions on the conservation of the Chinese white dolphin. (Line 153)
Reviewer 3 Report
Manuscript comments: fishes-1811410 “The potential of designating offshore wind farm to act as marine protected area – Conservation of the Chinese white dolphin regarding fishers’ perception”
General comments:
Dear Editor, after reading the manuscript with much attention and interest, I allow myself to point out the following. The manuscript reports the findings of explore the attitude and perception of the fishers towards the conservation of Chinese white dolphin and MPA to understand the potential compatibility of designating offshore wind farm as MPA. The manuscript is generally well structured, clearly describes the objective of the study, materials and methods are generally well described. The results are presented correctly, the use of tables and figures is adequate. The bibliographic review that the authors did to support their frame of reference is adequate. I have enjoyed reading the manuscript from beginning to end, the results found by the authors are very interesting and contribute to the knowledge regarding species that have interesting biological characteristics for conservation and food production purposes. As well as the role played by marine protected areas for these purposes. The only recommendation I could make to the authors is that they describe a little better what is related to the activity index in the materials and methods section, the actual form is a bit vague.
Although my native language is Spanish, it is evident that the document in its current form must be reviewed by an editorial service specialized in the English language.
Author Response
The authors greatly appreciate the thoughtful comments and suggestions.
1. We make a clear description in the research method section.
2. The revised manuscript has been carefully revised by two highly qualified native English speakers to improve the grammar, punctuation, spelling, and overall style.
.
Reviewer 4 Report
The authors outline a valiant effort to study to explore the attitude and perception of the fishers towards the conservation of Chinese white dolphin and MPA in order to understand the potential compatibility of designating offshore wind farm as MPA.
I found the ms well written, and I consider the basic premise, method developed and questions asked by this study as valuable and interesting. In general, the methodological approach and the statistical methods used for this study are valid and correctly applied to the data. Data interpretation are coherent and well presented. Tables and figures are also well presented and titled. The discussion is detailed enough and the scope of the ms very well presented with high quality of reference and examples.
Some minor comments
Introduction
In Lines 56 to 79, I would like to disaggregate the references mentioned in line 57 [12, 14-24] and to include each of these references to specific parts of the lines 58-79.
Results
Please clarify the lines 142-143 and what is "overall survey response rate"? Is this the cover ratio of the participated fishers? If not, which is the % total coverage of the participated fishers?
Author Response
The authors greatly appreciate the thoughtful comments and suggestions.
- All the references were all divided into four specific parts. (Line 54-73)
- The overall survey response rate is 76.7% (69/90). (Line 134).
Round 2
Reviewer 1 Report
The present manuscript is enhanced with respect to the original version. Yet, the most critical points are still present.
First, I firmly believe that an area devoted to industrial development (in the present case electricity generation) cannot act as a marine protected area, because framed within different policies, implemented by different ministries, voted to different goals, and managed for different objectives. Development areas (like wind-mill farms) can be contained within a larger MPA if judged a compatible use within a certain typology of zone. But this does not mean that wind-mill farms can act as marine protected areas. What is meant is that wind-mills for electricity generation represent an use compatible with a given typology of zone. The difference is substantial. For example ports exist within many MPAs, but this does not mean that ports can act as MPAs.
Second, there is still substantial bias in the focus of the manuscript. For example table 5 reports the frequencies of agreement to 7 different questions, of which 6 represent agreement to restrictions to fishing, and only 1 implies disagreement to those restrictions. Even if people would answer at random, people agreeing restrictions should be expected to be six times more numerous than those opposing them, because the former group is represented in 6 occasions while the second group is in one single question. There would be at least three different ways to perform this research correctly:
(1) Present only one question representing fishing restrictions and only one question representing disagreement to the proposed fishing restrictions.
(2) Present 6 pairs of questions, each pair representing agreement and disagreement with any of the proposed management actions.
(3) Give weights to the answers, in such a way that votes for the first 6 questions are multiplied by 1/6=0.16, and the last question representing disagreement is multiplied by 1/1=1.
Even with the present asymmetry of probabilities of positive and negative answers, it happens that the last option (disagreement) is the most frequent answer. This result is however obscured by summing up the six upper options (agreement). Why authors provide such a biased lecture?
Yet another issue with Table 5 is that the sum of frequencies equals to 90, while the reader is informed that the sample size (n) is 69. To make things more confused, percentages do sum up to 100%.
Third, almost every initiative at sea will find some favorable academics and others opposing to it. This is because there is a lot of guessing, and great paucity of sound evidence, possibly due to the inherent difficulties of experimental work at sea. Also, because many emerging topics are simply unknown. Wind-mill farms a re relatively new and we lack sound knowledge about their potential impacts. Among other things, we can see only immediate responses (e.g. colonization by sessile fauna), but known almost nothing about long-term, chronic effects (e.g. exposition to underwater noise and electromagnetic fields). We can however, learn something from analogous experiences. For example, there is substantial bibliography on the positive and negative impacts of dismissed oil platforms, as well as of artificial reefs. After tens of years, the prevailing opinion is that the effects of these interventions are case-sensitive, and it is difficult to anticipate the outcome of any given initiative at any given place. Since substantial uncertainty surrounds the present issue of possible effects of wind-mill farms on an endangered cetacean, it would be possibly wise to be cautious when writing from the academic standpoint. After all, it is often expected that academia output inform policy- and decision-making. The present version is substantially enhanced with respect to this issue, but I still find it unbalanced towards an unverified optimism.
Some minor comments follow:
Ln. Ln. 15: Change "specie" with "species".
Ln. 16: Add "must take into account" before "fishers perceptions".
Ln. 20: Change "The results show that" by "in order to assess that..." and link this sentenced to the previous one.
Ln. 21: Delete "Designating" and "to act as MPAs".
Lns. 22-23: Delete "all stakeholders... and ecosystem".
Ln. 23: Change "makes feasible" with "contains".
Ln. 24: Delete "on how".
Ln. 26: Change "Chinese white dolphin; Marine protected area; Offshore wind farm" with adequate keywords since these are already contained in the title, and title, keywords and abstract will be all used for indexing purposes by bibliographic databases.
Ln. 30: Insert a comma after "Taiwan Strait".
Ln. 31: Insert a comma after "monsoons”.
Ln. 41: Delete "marine mammal, the".
Ln. 44: Change "on" with "in".
Ln. 68: Change "bycatch" with "incidental catch". Bycatch are species not primarily target by the fishery, yet marketable. Incidental catches refer to unwanted species that are not marketable.
Ln. 69: Change "Their incidental... leads to" with leading to" and link this sentence to the previous one.
Ln. 71: Change "indirectly" with "contribute to"; "threaten" with “threat"; and "their" with "Chinese withe dolphin".
Ln. 73: Delete "the" before "economy"; change " create more job opportunities" with "job".
Ln. 75: Delete "such"; change "developments" with "development"; change greater" with "great"; delete "the" after "to".
Ln. 80: I suppose that "activities" have not been designated as MPAs. Possibly "places devoted to different activities" or the like?
Lns. 88-90: The reef effect also encompasses negative outcomes and the correct approach is to illustrate both. regarding the missing negative ones, consider e.g. higher detectability of sought species, attraction and concentration of large predatory fishes, increased fishing mortality due to concentration and enhanced detectability of target fishes, defaunation of the surrounding bottoms (halos) due to the concentration of predation in and around the structures, etc...
Ln. 90: As stated earlier and above, the idea that offshore wind farms can act as MPAs is wrong. Please delete this idea from the present manuscript. This does not imply that offshore wind farms cannot be placed within wider MPAs.
Ln. 91: The idea that offshore wind farms restrict fishing activities is right, but not good. Fishers were there well before wind-mills, and this make fishers legitimated to continue fishing in their traditional fishing grounds. The eventual displacement of traditional users of a given marine territory in favor of new economically and/or politically powerful actors constitutes a classical example of "ocean grabbing" (sensu Bennett et al., 2015).
Lns. 93-97: As actually written, it sounds like wind farm project developers buying consensus from fishers in order to convince them to stop being fishers for the sake of "local economic benefit". Authors are invited to assess what the wide local society think about loosing local fishers along with their cultural heritage, ecological knowledge, and contribution to the identity of the local coastal communities.
Lns. 99-100: As stated earlier above "designating offshore wind farms as MPAs" is conceptually wrong because inspired by different policies, aimed at different goals, and managed for distinct objectives.
Ln. 108: Change “motivated … society” with “was acceptable by all stakeholders”.
Ln. 109: Delete the subsentence between brackets. It is misleading and not necessary. Many of those poorly educated people hold relevant and accurate knowledge about that piece of sea, the Chinese white dolphin and the local fisheries. We have a lot to learn from them.
Ln. 111: Change “returns” with “returned”; please specify which other methods.
Ln. 112: Change “useful and available information” with “a complete illustration of the study, its objectives, and the way in which the provided information will serve the study purpose” or similar. This is standard practice in working out interviews. It is also required an statement of compliance with national laws on personal data and privacy.
Lns. 117-119: The purpose of the t-tests should be better illustrated. It is always the case that fishers opinions depend on their personal backgrounds. What is being tested for? What is the relevant hypothesis?
Ln. 119: Change “items” with “entries”, and “is” with “was”.
Ln. 121: Change “fishers” with “fisher”; and delete “their” because it is supposed that the questionnaire is individual.
Lns. 123 and 125: Change “Fishers” with “Fisher”.
Ln. 131: Change “are” with “were”; and “of” with “with”.
Ln. 133: Delete “the” before “advantageous”.
Ln. 137: Change “ratios” with “ratio”; delete “men and”; change “are” with “was”, and “have” with “had”.
Ln. 139: Insert “, respectively,” before “the major…”.
Ln. 140: Change “vessels” with “vessel”; insert “the most common” before “fishing”; change “technique” with “techniques”, and “in Yunin” with “of the Yunin”.
Ln. 143: Table 1: Please provide the extended name of “PVP”.
Ln. 146: Insert “that” after “think”.
Ln. 148: Change “two” with “both fishers and Chinese white dolphins”, change “resource” with “resources”.
Ln. 151: Add “by fishers” before “from”.
Ln. 153: Insert “incidentally” before “caught” and delete “as bycatch”.
Ln. 154: Change “Bycatch” with “Incidental catch” in Table 2.
Ln. 157-158: Delete “not the conservation … rather”.
Ln. 159: Change “are” with “were”.
Lns. 179-182: The structure and lecture of table 5 here provided is biased. Regarding structure, you must provide an equal number of positive and negative options by either (1) collapsing the first 6 questions to 1 stating e.g. “I do support management actions”; or (2) expand the last option to 6 statements, each of them negating (I do not support…) one and only one of the proposed restrictions. In its present form, it would be concluded that fishers support management actions even if they would respond at random.
Regarding lecture, the highest percentage of fishers (n=27, 30%) do not support the proposed management actions. Other people support some, but not all the proposed actions. However, it would prove difficult, or even impossible, to come to an agreement on which specific management action to implement because not of the single proposed actions has an score higher to that of those opposing any action. Surprisingly, the reported frequencies sum up to 90, in contrast with the stated number of people interviewed (n=69).
Ln. 190: Change “and operating ranges… white dolphin” with “occurrence of Chinese white dolphin and its habitat” or similar.
Lns. 190-196 and Lns. 205-206: This sections needs clarification. I can not understand what is meant by “the mean value of fishers…”. What data are summarizing these means? What is the variance associated to the reported means? What are the values reported over the brackets in the column “significance” of table 6? Should they be t-values, can authors explain negative values?
Ln. 201: Place a comma after “opinions”; change “depended” with “depending”, and “for” with “to”.
Lns. 202-205: This sentence seems truncated. It possibly was linked to the previous one. Please check.
Lns. 208-209: The reported means must be tied to a measure of the associated error, e.g. mean ± standard deviation.
Ln. 213: I cannot understand the meaning of “were varied no significant difference”.
Lns. 213-215: This sentence is unclear, please rewrite.
Ln. 216: Delete “in the discussion and”.
Ln. 217: Change “of” with “about a” or “on a” before “Chinese”; delete “convenient”.
Ln. 218: Change “of management strategies… from government” with “in management institutions”.
Ln. 228: Change “intends” with “is intended”.
Ln. 231: Insert “a” before “MPA”.
Lns. 232-234: Delete “consider… strike a”; delete “between”.
Ln. 237: Place a comma after “designation”; change “can” with “which”.
Lns. 239-240: Insert “about” after “concerned”; and “the” before “MPA”.
Lns. 240-242: Provide adequate references for these assertions.
Ln. 243: Change “bycatch” with “incidental catch”.
Ln. 244: Change “cetacean” with “cetaceans”, “is” with “are”, “main catch” with “target”, and “it refers to” with “fishers claim that”.
Ln. 249: Change “the MPA” with “MPAs”, and “cetacean” with “cetaceans”.
Ln. 250: Insert “properly” or similar before “designed”; change designated” with “certain” of similar to avoid repetition within the same sentence.
Ln. 250: Insert “properly” before “designed”. Delete “the” before “habitats”.
Ln. 251: Delete “where” before “relate”, and “mainly”.
Ln. 253: Change “core zone” with “core zones”.
Ln. 254: Change “management… strictlyʺ with “enforcement to avoid”; insert “coastal” before “development”.
Ln. 255: Add “the” before “core”; change “zone” with “zones”.
Ln. 256: Change “for sustainable use” with “the sustainable exploitation”; “resource” with “resources”; and “local community of fisher” with “by the local fishers community”.
Lns. 256-258: Start the sentence as “MPA zoning plan would be a tool to recover fish resources for the benefit of Chinese white dolphin and local fishers, to promote a …”; place a comma after “environment”; insert ”protect” before “biodiversity”.
Ln. 264: Delete “Through” and start the sentence at “Incorporation…”.
Ln. 265-256: Change “can improve the” with “promotes”; and “management strategies” with “acceptance and enhance compliance”.
Lns. 270-272: Change “and acceptable information or data” with “information”, and “convince them that “ with “get their support to locate an MPA of appropriate size”.
Ln. 280: Change “fishery“ with “fishing“, and “degeneration” with “overexploitation”.
Lns. 281-282: Insert “degradation of the” before “marine environment”; change “are” with “were possibly”, and “law enforcement of regulation” with “law and regulation enforcement”.
Ln. 284: Change “have sufficient legal force” with “provide sufficient legal provisions”, and “MPA2 with “MPAs”.
Ln. 285: Change “fishing gear” with “fishing gears”.
Ln. 286: Change “method” with “methods”; and “it may be able to sustain…” with “then they also contribute to the recovery of the Chinese white dolphin population”.
Ln. 287-288: Change “low management effectiveness, and then” with “management effectiveness is low, hence”; insert “that” before “existing”; change “may” with “will”; and add “to run as usual” after “may continue”.
Ln. 289: Insert “a” before “MPA”.
Ln. 291: Change “on illegal” with “to”. Illegal fishing is prohibited rather than limited.
Ln. 292: Change “reserved area” with “the reserve area”, and “the integration of cross department” with “cross-integration of departments”.
Ln. 294: Change “in terms of jurisdiction and” with “for”.
Ln. 295: Rewrite as “effective management and enforcement of the MPA”.
Ln. 297: Start the sentence as “A previous…”; delete “that”.
Ln. 300: Rewrite as “an offshore windfarm incorporated within a newly designated MPA”.
Lns. 303-304: Add “to a certain degree and” after “complementary”.
Lns. 304-305: I suggest rewriting as e.g. “Placing wind farms adjacent or within wider MPAs can provide reciprocal benefit in some circumstances”.
Lns. 306-307: Change “wind farm” with “wind farms”, “designated” with “designation of”, “is advancing” with “advances”; and “fisher” with “fishers”.
Ln. 309: Change “area” with “areas”.
Ln. 310: Delete “on”; change “by” with “of the”.
Ln. 312: Change “on social and economic consequences by” with “social and economic consequences of the”.
Ln. 312: Change “area of” with “areas dedicated to”, and “farm” with “farms”.
Ln. 335: Delete “, and associated local fishing”.
Ln. 339: Delete “these”, and “platform has” with “platforms have”.
Ln. 341: Change “as exclusive fishing zone” with “fishery exclusion area”, since many types of fishing, or even all fishing at all, is usually forbidden around oil platforms.
Ln. 343: Change “excluding” with “allowed”.
Lns. 345-349: The tree sentences “Therefore, local fisher… by fishing activities” must be deleted because they are not supported by facts, neither by adequate references. In addition, I think that they are simply wrong.
First, overfishing is a problem difficult to tackle because rebuilding fished stocks is often difficult, sometimes impossible, takes years, ask for substantial reduction of fishing effort, which in turn imply economic and social sacrifices. To provide an easy formula to get there with little effort is a fallacy.
Second, control of the harvest is very difficult. External observers are useful but expensive, while self-reporting is inconsistent. It is unveiled in which way the presence of wind-mills would help to control the harvest.
Third, the fishing gear is more susceptible of being damaged than the wind turbine infrastructure.
Nevertheless, some type of fishing (e.g. vertical angling) can possibly be carried out within the wind-mill farm. Providing traditional fishers with rights to fish compatibly with wind-mill integrity can be a workable option. However, in order to be economically viable, it would possibly needed that those fishing rights are exclusively given to those professional fishers actually working in the area where the wind-mill farm is intended to be developed, at least at the initial phase. The wind-mill farm can also be designed in order to have areas where those fishers can deploy other types of compatible fishing gear like traps. It should be kept in mind that there would be an initial period in which low catches will not compensate for shifting fishing gear, so that some type of compensation or support could be needed.
Ln. 351: Change “multiple” with “different”.
Ln. 354: Add “the” before “local”; delete “The” before” “Economic”; change “associate” with “associated”.
Ln. 355-356: Change “in the aspects of” with “regarding” or similar.
Ln. 356-357: It is unveiled how the proposed economic benefits would promote ecosystem sustainability, and no reference is provided to support this assertion.
Ln. 359: Change “eliminate” with “be overcome”; delete the sentence “Designating… circumstance” since is not supported by experimental data, nor literature references.
Lns. 360-362: Insert “proposed” before “offshore wind farm”; delete “MPA and”; change “tools” with “tool”; change “and the ecosystem” with “, the ecosystem, and the local society”.
Ln. 373: Change “eduction” with “education”.
Ln. 375: I will left “charter fishing” out of the present formula, since the ethically correct approach would be to ensure first the viability of the professional fishers traditionally working there and therefore being negatively affected by the the wind-mill farm. See also the provided reference about ocean grabbing.
Ln. 377: Delete “The”.
Ln. 380: Delete “the” before “ecological”.
Ln. 384: Change “has overlapped” with “overlaps”, and “fisher” with “fishers”.
Ln. 385: Change “applicable” with “the application of”, “may” with “would”, and “their” with “fishers”.
Ln. 386: Insert “their” before “opposition”.
Cited reference: Bennett NJ, Govan H, Satterfield T (2015). Ocean grabbing. Marine Policy 57 (61-68).
Author Response
Thank you for your precious comments and advice. Please see the attachment.

Reviewer 2 Report
I have made many editorial changes on the document

Author Response
The authors greatly appreciate the thoughtful suggestions.
- Thank you very much for finding these errors. We are sorry for these english problems and have corrected them according to your suggestion.
- yes, we mean marine ecosystems in this sentence.
- yes, it is a degree.
- Thank you very much for finding these errors. The authors have carefully reviewed and corrected all references.
Round 3
Reviewer 1 Report
The present manuscript has been enhanced with respect to the previous versions. There are a number of suggestions from my side that have not been taken into consideration by authors, but these can be retained a matter of personal style.
Major points have been addressed, and the present version is quite satisfactory. I take the present opportunity to indicate a small number of corrections, which in my view could help enhancing the resulting manuscript.
Ln. 16: Change "an MPA" with "a MPA".
Ln. 22: Change "all" with "different".
Ln. 42: Add "dolphin" or "species" after "This".
Ln. 51: Change "can be considered" with "is".
Ln. 63: Change "damages" with "is released in".
Lns. 74-75: Change "been increasing" with "increased".
Ln. 93: The term "eliminating" is too optimistic, in my view. I suggest "ameliorating", which is commonly used in most analogous sentences about conflicts.
Ln. 96: Change to be an effective tool in the" with "to bring the opportunity for the" or similar.
Lns. 96-97: Delete "the improvement"; and also "the" before "provision".
Ln. 99: Change "better understanding" with "defining" or similar.
Ln. 178: Change "were satisfied" with "are willing to"; insert "one or another" before "fishery management"; and change "measures" with "measure".
Lns. 202-203: Table 6. Change "Significance (p<0.05)" with "associated p-value" as header of the last column. If I understood correctly the comment provided by authors, the numbers over the brackets in the last column are values of the t-statistic. If so, it must be specified in the column header.
Ln. 222: The correct term is "incidental catch" instead of "bycatch". The second is the capture of species different from the target one(s) but marketable anyway. In contrast, incidental catches of protected species cannot be marketed.
Ln. 225: Change "main catch" with "target".
Ln. 239: Change "biodiverse" with "diverse"; and "environment" with "ecosystem" because resilience and biodiversity are attributes of ecosystems, not environments.
Ln. 243: Inset "allowed" before "fishing gear".
Ln. 663: I intend "legal provisions" or "legal instruments" instead of "legal force".
Ln. 286: Start sentence as "If improperly implemented, it can rob...". The implementation of mechanism of social justice during the process can avoid such possibility.
Lns. 299-311: This passage describes the positive effects. There are also negative ones, which I indicated in previous reviews. It is my opinion that it would be fair to indicate both, positive and negative ones.
Ln. 328: Delete the underscore "_" between "up" and "with".
Ln. 370: Add "Chinese white dolphin" before "food resources".
Ln. 383: Change "addition" with "institution" or "implementation" or the like.
Author Response
all authors are grateful for the suggestion from reviewer 1. The manuscript has been revised, please see the attachment
Best wishes,
